# Knowledge, Perceptions, and Practices of Traffic Police Officers Towards Air Pollution in Addis Ababa, Ethiopia: An Exploratory Study

**DOI:** 10.3390/ijerph23010060

**Published:** 2025-12-31

**Authors:** Andualem Ayele, Andualem Mekonnen, Eyale Bayable, Marc N. Fiddler, George Stone, Solomon Bililign

**Affiliations:** 1Center for Environmental Science, Addis Ababa University, Addis Ababa P.O. Box 1176, Ethiopia; andualem.ayele@ethernet.edu.et (A.A.);; 2Department of Chemistry, North Carolina Agricultural and Technical State University, Greensboro, NC 27411, USA; 3Department of Marketing and Supply Chain Management, North Carolina Agricultural and Technical State University, Greensboro, NC 27411, USA; 4Department of Physics, North Carolina Agricultural and Technical State University, Greensboro, NC 27411, USA; 5Institute of Geophysics, Space Science and Astronomy, Addis Ababa University, Addis Ababa P.O. Box 1176, Ethiopia

**Keywords:** perception, practices, traffic police, occupational health, air pollution, Addis Ababa

## Abstract

Traffic police officers represent a critical occupational group with high vulnerability to vehicular air pollution, a severe environmental health threat in rapidly urbanizing metropolises such as Addis Ababa. This cross-sectional study explored occupational exposure, protective practices, health risks, perceptions, and awareness of air-quality-associated health risks among 120 traffic police officers in Addis Ababa. The officers were mostly male (80%) and married (93.3%), with the majority (62.6%) having served for more than ten years. While vehicle emissions were consistently recognized as the main source of air pollution, critical knowledge gaps were identified, i.e., only 24.2% had received pollution-related training, fewer than half (45.8%) were aware of government policies, and just 9.2% reported collaboration with environmental authorities. Awareness of the Air Quality Index (AQI) was generally low, and regular monitoring of AQI was limited. Self-reported health symptoms were highly prevalent among participants, with cough (75.0%), eye irritation (61.7%), sneezing (58.3%), and runny nose (55.8%) being the most frequently reported. Notably, sneezing, runny nose, eye irritation, and psychological stress showed significant association with perceived pollution levels at the workplace (*p* < 0.05), while blood pressure, cough, difficulty concentrating, and sleep loss were not significantly associated (*p* > 0.05). A higher prevalence of symptoms was generally observed in groups experiencing moderate-to-very high levels of pollution. Protective measures were applied inconsistently; while 63.3% of participants reported using masks, their beliefs about the effectiveness of using masks varied. Relocation (60%) and use of face covers/glasses (13.3%) were less commonly practiced. Overall, traffic police officers are exposed to occupational air pollution, which is associated with various health symptoms. These findings highlight the need for enhanced training, clearer communication of policies, stronger institutional engagement, the provision of standardized protective masks, and the promotion of AQI utilization to reduce occupational health risks and safeguard the wellbeing of traffic police officers in Addis Ababa.

## 1. Introduction

Air pollution is one of the most serious global environmental health threats, contributing to millions of deaths annually, and it affects nearly every organ system [1]. It is the second leading global environmental risk factor for death, responsible for approximately 8.1 million deaths worldwide in 2021, including over 700,000 children under the age of five [1]. More recently, estimates from the state of Global Air 2025 [2] report indicate that air pollution caused about 7.9 million deaths in 2023, with nearly 86% of these deaths linked to non-communicated diseases such as cardiovascular and respiratory conditions. During this period, an estimated 4.9 million deaths were attributable to PM_2.5_ (particulate matter with an aerodynamic diameter of 2.5 μm or smaller) exposure, 2.8 million to household air pollution, and approximately 470,000 to ozone exposure. In 2019, air pollution caused an estimated 1.1 million deaths, with 394,000 attributable to ambient pollution across Africa [3]. Air pollution, including ambient and household air pollution, is the second leading risk factor for deaths in East Africa, accounting for 294,000 deaths in 2021 [4].

In Ethiopia, household air pollution causes nearly 39,000 deaths annually and results in economic losses of about ETB 281 billion (USD 8 billion), while ambient air pollution levels—nearly five times higher than the WHO 2021 guideline—contribute to over 25,000 additional deaths each year and impose a further ETB 91 billion (USD 2.6 billion) in health-related economic losses [5,6].

Addis Ababa’s population growth has resulted in a rapid urbanization and a significant increase in per capita income, leading to an escalating demand for energy and transportation [7]. Although Addis Ababa possesses an extensive road network consisting of asphalt, gravel, and cobblestone surfaces, the existing road network is inadequate to address the city’s increasing transportation needs, particularly along major corridors. Traffic congestion is one of the urbanization challenges facing Addis Ababa City. For example, the number of registered vehicles in Addis Abba in 2020 reached 630,440. According to reports, the number has increased to over a million by 2024/2025 [8]. About half are gasoline-powered, and the other half are diesel-powered, and fewer than 10,000 are electric vehicles. A considerable proportion of the city’s vehicle fleet is reported to be old and poorly maintained [9], and this aging vehicle fleet is one of the major contributors to high pollution in Addis Ababa [5,9,10,11]. Traffic police officers, who are consistently stationed at highly congested intersections for prolonged periods, are thus among the most affected occupational groups, since they face direct and sustained exposure to vehicular emissions.

Air pollutants are the major source of urban air quality problems, contributing to health risks and environmental degradation. The main sources of pollutants are incomplete combustion from diesel vehicles, dust from unpaved roads and tires, biomass burning, construction, and industrial activities [12,13,14,15]. Tire wear is an increasingly recognized source of air pollution in urban areas with high traffic volume [13,16]. The friction between tires and the road surface emits a range of particles such as microplastics, rubber fragments, and heavy metals that contribute PM_2.5_ [10]. Brake pad wear creates a non-exhaust emission of pollution, which includes ultrafine particles and volatile organic compounds [17,18]. The primary air pollutants from vehicles are carbon monoxide (CO), nitrogen oxides (NOx), black carbon (BC), benzene, and other aromatic compounds such as polycyclic aromatic hydrocarbons (PAHs) [10,19]. These ultra-fine pollutants can penetrate deep into the lungs and bloodstream, leading to severe health outcomes including asthma, Chronic Obstructive Pulmonary Disease (COPD), lung cancer, pneumonia, ischemic heart disease, hypertension, heart attack, and stroke [20,21,22,23,24,25,26,27,28,29]. At-risk individuals exposed to the elevated levels of air pollution due to their occupational activities include traffic police, street vendors, taxi and public transport drivers, construction workers, and residents near congested areas.

Traffic police officers are consistently stationed at busy traffic intersections and typically work over 8 h per day outdoors as part of their job. This makes them among the most affected occupational groups, facing direct and sustained exposure to vehicular emissions and constant exposure to air pollutants, resulting in an elevated rate of respiratory and cardiovascular diseases [24,30,31,32,33,34].

These conditions highlight the need to assess traffic police officers’ knowledge, perception, and practices (KPP) regarding air pollution exposure. Understanding the KPP of high-risk groups is a critical step in designing effective public health interventions. A study of traffic police in Nairobi, Kenya, found that, while a significant proportion of officers (40.2%) self-rated their knowledge of motor vehicle pollution as ‘good’ or ‘very good’, there was a stark disconnect between this knowledge and practical mitigation [35]. Crucially, a majority felt the police administration was not taking adequate measures to protect them, and most did not use personal protective equipment. This highlights a critical gap between knowledge and action. Our study builds upon this by assessing KPP among traffic police in Addis Ababa. A separate publication to determine the correlation between these findings with direct, objective measurements of black carbon exposure will provide a more comprehensive understanding of the barriers to implementing protective practices.

KPP frameworks have been widely applied to occupational and environmental health research to identify gaps between awareness and protective actions. Globally, KPP studies among traffic police and other high-risk occupational groups reveal that even when knowledge is high, perceptions of risks are often underestimated, and protective practice are inconsistently applied [35,36]. In urban African settings, this disconnect is exacerbated by structural and administrative limitations, such as a lack of personal protection equipment (PPE) provisioning, insufficient traffic management, and poor enforcement of pollution control measure. Therefore, evaluating KPP among traffic police officers in Addis Ababa is critical to inform targeted occupational health interventions, public awareness campaigns, and policy measures aimed at mitigating exposure.

In addition to occupational health concerns, traffic police officers’ prolonged exposure has broader public health implications, as they often serve as intermediaries between authorities and the public in traffic management and environmental regulation enforcement. Their knowledge and perception can influence public adherence to traffic and pollution control measures, creating a dual importance for KPP assessment: protecting the officers themselves and enhancing city-wide air quality interventions.

Several studies conducted in Addis Ababa have documented elevated concentrations of ambient particulate matter and their associated health and economic impacts. PM_2.5_ monitoring and chemical characterization studies have shown that concentrations frequently exceed World Health Organization guideline values and are largely influenced by traffic emissions, biomass burning, and road dust [37,38]. Health and economic burden analyses further demonstrate substantial mortality and economic losses attributable to PM_2.5_ exposure in the city [39]. In addition, assessments of PM_2.5_ and PM_10_ across indoor and roadside micro-environments indicate that roadside environments pose significant health risks due to long-term particulate matter exposure [40].

Despite this evidence, existing studies in Addis Ababa are largely limited to environmental measurements and health impact estimation and do not assess knowledge, perception, or preventive practices related to air pollution among highly exposed occupational groups such as traffic police, underscoring a critical gap addressed by the present study. This study seeks to answer the following questions: (1) what is the level of awareness of traffic rules and regulations among traffic police officers? (2) How do traffic police officers perceive the challenges and risks associated with their duties? (3) What is the most effective way to disseminate educational materials on impacts of air pollution to the public?

## 2. Materials and Methods

### 2.1. Description of Study Area

Addis Ababa, the capital of Ethiopia and the seat of the African Union, Economic Commission for Africa, Africa Centre for Disease control, and several international organizations, also serves as the country’s political, commercial, diplomatic, and cultural hub. Addis Ababa is a chartered city with a three-tier administrative structure consisting of the city administration, sub-cities, and woredas (Figure 1). The city is organized into eleven sub-cities, which serve as the second level of administration beneath the city administration. In terms of land area, Bole is the largest sub-city, followed by Akaki-Kality and Yeka, Addis Ketema, Lideta, and Arada in order of decreasing size. The newest sub-city, Lemi Kura, was established by separating areas from Bole and Yeka. Each sub-city is further divided into woredas, the smallest administrative units of the city.

According to the Ethiopian Statistics Service (ESS), the projected population of Addis Ababa in July 2024 was 4,030,000, consisting of 1,900,000 males and 2,130,000 females [41].

Most of the city lies at the foot of the Entoto mountain range, with elevations ranging from approximately 2900 m in the north down to 2300 m in the southern periphery near the Akaki Plains [42]. This unique topography significantly influences local air circulation, as the surrounding mountains can trap atmospheric pollutants within the valley, limiting their dispersion by wind [43,44,45]. The city experiences a subtropical highland climate, with temperatures ranging from mild to cool. The long-term mean annual maximum and minimum temperatures of Addis Ababa are 25.8 °C (78.4 °F) and 12.6 °C (54.7 °F), respectively. Over the past few decades, the maximum temperature has increased by about 2.7 °C (4.9 °F), while the minimum temperature shows smaller variations [46].

### 2.2. Study Design and Data Collection

A cross-sectional study was conducted to assess the KPP regarding air pollution among traffic police officers in Addis Ababa, Ethiopia, between May and June 2025. Due to security constraints (not publishing the exact number of traffic police in the city), data could not be collected from the entire population of officers. Instead, 120 officers were purposely sampled from six sub-cities, with 20 participants per sub-city (10 from the morning shift and 10 from the afternoon shift). Within each shift, officers were randomly approached during daily orientation sessions prior to deployment to high-traffic intersections.

Data were collected using a 40-item structured questionnaire covering demographic characteristics, awareness, knowledge, perception, and practice regarding occupational air pollution. This was supplemented with field observations to cross-validate self-reported practices, providing a richer and more reliable assessment of actual occupational behaviors.

To enhance the generalizability of our findings, post-stratification weighting was applied to adjust for differences in age and gender distribution across the sample [47]. This approach allowed us to extrapolate findings to the broader population of traffic officers in Addis Ababa, accounting for potential sampling bias [46,47].

### 2.3. Statistical Analysis

Data were coded and analyzed using IBM SPSS Statistics version 25 and Microsoft Excel 2016. Descriptive statistics were used to summarize demographic characteristics and KPP response. To enhance analytical depth, subgroup analyses were conducted to examine variation in KPP across demographic and occupational characteristics, including years of service, educational level, and work location. This analysis provided insight into differential awareness and protective behaviors among traffic police officer with varying exposure. To examine the relationship between self-reported pollution exposure and the prevalence of health symptoms, cross-tabulations and chi-square tests were performed. The results were presented in tables, highlighting patterns of symptom prevalence across different exposure levels.

We used post-stratification weighting to adjust the survey sample for differences in age distribution and then extrapolated the weighted results to estimate health risk awareness in the adult population of Addis Ababa [47]. A statistical adjustment method called post-stratification was used to match survey samples to the target population’s known characteristics. Following data collection, it entails allocating weights to survey participants according to the distribution of demographic factors, such age, gender, or geography. Post-stratification is primarily used to increase survey estimate accuracy, account for any sample biases, and enable results to be extrapolated to the target population using Equation (1) [47,48].(1)$$W_{i}=\frac{P_{i}}{S_{I}}$$
where W_i_ is the weight for the stratum, P_i_ is the proportion of the population in stratum i (obtained from censes or official demography data), and S_i_ is the proportional sample in stratum i.

To generalize the survey findings to the adult population of Addis Ababa (≥18 years), population counts by 5-year age bands were obtained from the most recent census. These were re-mapped to match the survey age categories: 18–25, 26–35, 36–45, and ≥46 years. For example, ages 18–19 were estimated as 40% of the 15–19 age group and age 25 as 20% of the 25–29 group. Weights for each age group were calculated as the proportion of the population in that group divided by the proportion of the sample in that group. These weights were then applied to all health outcome variables to produce estimates representative of the adult population of Addis Ababa.

## 3. Results

As noted, the respondents for this study were traffic police officers, a group directly exposed to and actively involved in managing and monitoring road traffic—the primary source of urban air pollution. Their occupational role placed them at the forefront of enforcing traffic regulations, controlling vehicle flow, and interacting with the public—an occupation that positions them as both influencers of, and stakeholders in, the effort to reduce air pollution.

### 3.1. Demographics

Table 1 presents the demographic characteristics of the respondents. A total of 120 traffic police officer were included in the study, representing six sub-cities of Addis Ababa. Of these, 80% (*n* = 96) were male and 20% (*n* = 24) were female. The majority were married (93.3%, *n* = 112), while 5.8% (*n* = 7) were single and 0.9% (*n* = 1) were divorced. The age distribution showed that nearly half (48.3%, *n* = 58) were between 36 and 45 years old, 38.3% (*n* = 46) were aged 26–35 years, 8.3% (*n* = 10) were aged 66 and above, and 5% (*n* = 6) were aged 18–25 years.

With respect to work experience, 62.6% of the respondents reported having more than 10 years of service, while 32.5% had 6–10 years of experience. Only 5% had worked for 1–5 years. Educational attainment varied among respondents. The largest proportion reported holding a diploma (45.8%, *n* = 55), followed by grade 12 completion (22.5%, *n* = 27), and a bachelor’s degree (22.5%, *n* = 27). A smaller proportion held an MSc (4.2%, *n* = 5). Six respondents (5.0%) indicated that their educational level was grade 10. In this context, “diploma” refers to a post-secondary qualification below the level of a bachelor’s degree. All participants reported that they did not smoke cigarettes. This is due to cigarette smoking being a restricted behavior for occupational ethics rules and regulations, making non-smoking the behavioral norm. While private tobacco use is possible, despite the reported rate, use in the general Ethiopian population is low, with only 5.0% using any tobacco product and 3.3% being smokers [49]. In terms of alcohol use, 17.5% (*n* = 21) reported drinking alcohol, whereas 82.5% (*n* = 99) reported abstaining from the use of alcohol. Among those who drank alcohol, 1.7% (*n* = 2) consumed alcohol frequently and 15.8% (*n* = 19) consumed it occasionally. The remaining 82.5% (*n* = 99) did not answer the question on frequency of drinking because they were non-drinkers. The demographic profile shows that most respondents were middle-aged men with extensive service experience, suggesting a workforce with considerable exposure to traffic-related occupational hazards over time. While such experience may contribute to a better understanding of traffic management, it may also mean prolonged exposure to vehicular emissions and associated health risks.

### 3.2. Descriptive Summary of Health Symptoms

The health symptoms reported by the traffic police officers exposed to air pollution are summarized in Table 2. The findings reveal that respiratory-related symptoms such as cough (75.0%), eye irritation (61.7%), sneezing (58.3%), and runny nose (55.8%), were the most common and most reported health-related side-effects associated with the job. A substantial proportion of the participants reported experiencing these symptoms, even if only rarely. The zero rate of cigarette smoking suggests that the observed respiratory symptoms are more likely related to occupational or household exposure rather than personal smoking behavior. While we acknowledge that having no smokers in this study may limit the applicability to the general public, the rate of smoking is quite low (only 3.3% in Ethiopia), though local rates may vary [49].

In contrast to physical symptoms, physiological complaints such as blood pressure issues (17.5%) and difficulties in concentration (15.0%) were less prevalent. Psychological stress (33.3%) and sleep loss (26.7%) were also noted, indicating that beyond physical discomfort, participants experienced some psychosocial impact as well.

These results suggest that environmental or lifestyle-related exposures may predominantly affect the respiratory system, with secondary effects on psychological well-being and sleep quality as reported. The relatively high prevalence of stress and sleep loss highlights the need for interventions that address not only physical health but also mental health outcomes. Public health measures and workplace or community-based strategies could therefore benefit from incorporating both preventive and supportive approaches to mitigate respiratory irritation while also promoting stress management and healthy sleep practices. These findings indicate that traffic police officers face a variety of physical and psychological symptoms, many of which align with known short- and long-term health effects of urban air pollution exposure.

### 3.3. Knowledge and Awareness of Air Pollution

We assessed traffic police officers’ knowledge of air pollution sources using a multiple-response survey across different educational levels, with the distribution of perceived contributors shown in Figure 2. Across all educational levels, vehicle emissions were consistently identified as the primary contributor, with 73.9% of grade 12 and 38.5% of grade 10 respondents reporting it as a major source. In contrast, respondents with higher educational attainment (diploma, degree, MSc) prioritized industrial and construction-related pollution over vehicle emissions, indicating a more nuanced understanding of drivers of pollution sources.

Industrial pollution was predominantly recognized by individuals with advanced education, particularly MSC holders (42.9%) and degree holders (35.5%). This pattern suggests that higher educational levels may correlate with increased awareness of less visible or systemic sources of pollution, such as industrial activities. Similarly, construction-related pollution was most frequently selected by diploma holders (38.3%), which may reflect occupational or community exposure experiences.

Garbage- and cooking-related pollution were less prominently recognized across all educational groups, although diploma and degree holders still reported relatively high awareness (25–28% for garbage and 11–13% for cooking). Interestingly, the perception of cooking as a source of pollution increased slightly among respondents with higher education (MSC: 14.3%), potentially reflecting knowledge of indoor air quality and health implications of household emissions.

These findings highlight the role of education in shaping environmental awareness. Individuals with lower educational levels tended to associate pollution with visible sources such as traffic emissions, whereas higher education levels corresponded with a more comprehensive understanding of industrial, construction, and indoor pollution sources.

Respondents’ awareness and engagement with the Air Quality Index (AQI) varied across educational levels (Figure 3). The results showed that awareness increased with educational level, but regular engagement remained limited. Among grade 10 officers, 16.67% were aware without understanding, 50% knew of AQI but did not follow it, and none monitored it regularly. Grade 12 officers showed slightly higher engagement, with 25.93% checking AQI occasionally. Diploma (equivalent to associate degree in community colleges) and degree (bachelor’s degree) holders demonstrated more consistent engagement, with 14–15% monitoring AQI regularly. MSc respondents expressed the highest lack of awareness or understanding (80% total), and none monitored AQI regularly. Interestingly, the MSc respondents deviate from this general pattern, showing the highest lack of awareness despite their educational attainment. Overall, the majority of participants were aware of AQI, although understanding and regular engagement were limited. This suggests that education level alone may not guarantee better understanding or engagement with AQI information.

The findings revealed a substantial perceived gap in training and police awareness. When questioned about training, only 24.2% (n = 29) of respondents reported having received training related to pollution, while 75.8% (n = 91) reported no such training (Table 3).

Awareness of government air pollution reduction policies related to vehicles was similarly limited, with 45.8% (*n* = 55) officer were familiar with government’s air pollution reduction policies, while the majority 54.2% (*n* = 65) reported they were unaware of the polices. This lack awareness was compounded by low level of collaboration with environmental authorities. Only 9.2% (*n* = 11) officers reported that their office engaging regularly, 37.5% (*n* = 45) reported that occasional collaboration, 24.2% (*n* = 29) no collaboration, and 29.2% (*n* = 35) did not know whether any collaborative mechanism even existed.

Regarding AQI, only 10% reported checking it regularly, while 83.3% were aware of its existence. These findings indicated that the general awareness is high for the presence of AQI but comprehension and active engagement are low.

These findings highlight significant gaps in training, policy awareness, and institutional engagement among traffic police officers. Limited structured training and low collaboration rates reduce their capacity to act as informed advocates for air pollution mitigation. Enhancing formal training programs, improving communication of government policies, and fostering regular collaboration with environmental authorities could strengthen officers’ role in occupational risk reduction and public education on air quality.

The survey assessed traffic police awareness of health risks associated with air pollution, allowing multiple responses. The distribution of perceived health risks across age groups is presented in Table 4. Overall, respiratory disease was the most frequently recognized risk (82 responses, 69.5%), followed by high blood pressure (50 responses, 42.4%), lung-related damage (36 responses, 30.5%), and heart-related disease (34 responses, 28.8%). Awareness varied by age group: the 36–45 years group showed the highest recognition across most risks, particularly respiratory problems (39 responses, 47.6%) and blood pressure (27 responses, 54.0%), while the 18–25 years group reported lower overall awareness. Among those 46 and older, respiratory problems (10 responses, 12.2%) were the most-identified risk, with limited recognition of heart and lung issues. Only one respondent indicated uncertainty. These results indicate that while respiratory risks are widely recognized, overall awareness of other air-pollution-related health risks among traffic police is moderate and varies with age.

### 3.4. Perception of Traffic Police on Air Pollution

This study investigates the perceived environmental risks faced by traffic police officers, with Table 5 presenting data on their perceptions regarding the evolution of Addis Ababa’s pollution, the micro-environmental conditions at their workplaces, and the associated health outcomes for further analysis. Most officers, 62 (51.7%), reported that the air quality had improved, while 35 (29.2%) believed it had worsened. Meanwhile, 15 respondents (12.5%) indicated no change, and 8 officers (6.7%) reported that they did not know. This implies the sample’s mixed perceptions of air quality in Addis Ababa, highlighting the need for strengthened air pollution communication, regular monitoring, and targeted awareness programs for officers. Regarding the health impact of air pollution, 38.3% of officers perceived the effects as severe and 37.5% as mild. Only 19.2% believed there were no health effects, and 5% reported that they did not know. This finding indicates that even with perceived improvement in air quality, a significant proportion of officers still experience a fear of health impacts. This discrepancy suggests that while there is a recognition of city-wide environmental initiatives, such as greening projects, road expansions, and promotion of cleaner transport, localized occupational exposure remains a serious concern. This highlights the need for targeted interventions and education for high-risk groups.

Regarding challenges to pollution reduction, respondents emphasized technical, institutional, social, and governance barriers (Table 5). The lack of advanced pollution-monitoring technology was most frequently cited (27.8% of responses; 57.5% of cases), highlighting perceived limitations in accurately assessing and managing air quality. Insufficient training and knowledge on environmental issues (22.6%; 46.7%) and inadequate legal support (20.6%; 42.5%) were also noted as significant institutional constraints. Social challenges, particularly low public cooperation (18.5%; 38.3%), and governance issues, including poor coordination among traffic, environmental, and transport departments (10.5%; 21.7%), were additionally recognized. These findings indicate that respondents perceive effective pollution reduction as dependent not only on technical and regulatory capacity but also on inter-agency coordination and public engagement.

Figure 4 illustrates the distribution of perceived causes of traffic-related air pollution by length of service among traffic police officers. Understanding how experience influences awareness can provide insights into knowledge formation within occupational roles that interact closely with environmental conditions. Among officers with 1–5 years of experience, old vehicles were identified as the primary contributor to air pollution (57.1%). This finding contrasts with the responses of more experienced officers, among whom the perception of old vehicles as the main cause decreased notably, with only 38.1% in the 6–10 years of service group and 38.5% in the group with over 10 years of service. This disparity suggests that early-career officers may rely more heavily on observable, tangible sources of pollution. Interestingly, these officers also reported a relatively high concern regarding lack of public awareness (28.6%), suggesting that young officers perceive driver or vehicles owners’ behavior as the result of a lack of awareness of how their everyday practices contribute to air pollution. Such practices include excessive idling, poor vehicle maintenance, and low use of green transport options. Young officers were also more likely to be frustrated by the non-compliant behaviors of drivers.

Conversely, the perception of vehicular congestion was more frequently reported by officers with greater experience. With only 14.3% of the 1–5-year group identified congestion as a major contributor, the percentage rose to 34.5% among those with 6–10 years of service and remained relatively high (27.3%) among those with over a decade of experience. A similar trend appears in the recognition of poor traffic light management as a contributor to air pollution. While none of the officers in the 1–5 years group identified this as a factor, 8.3% of the 6–10 years group and 10.5% of senior officers did. This gradual increase again suggests that a broader understanding of infrastructural inefficiencies develops with professional maturity and exposure. When data is aggregated across all age groups, the most frequently reported issue was old vehicles, followed by congestion, lack of public awareness, and poor traffic light management.

Overall, the findings suggest that longer service is associated with broader recognition of multiple pollution sources and underlying causes, with persistent concern about old vehicles and congestion. Interventions should prioritize phasing-out aging vehicles through stricter inspection regimes, emissions testing, and fleet renewal incentives. Congestion reduction strategies—including improved public transport, dynamic traffic management, and anti-idling enforcement—are also essential. Upgrading signal systems and building traffic management capacity could reduce infrastructure inefficiencies. Finally, public awareness campaigns, supported by strong enforcement and fuel quality standards, remain critical for long-term behavioral change and emission reduction.

Perceptions of major air pollution cause across age groups were analyzed using self-administered questionnaires data from traffic police officer, extrapolated to the general to the general population through post-stratification (Table 6). The results indicated that old vehicles were perceived as the most significant contributor, affecting 72.7% of the population. Congestion was also considered important, impacting 53.0%, particularly among the 26–35 age group. Poor traffic light management was selected least frequently, accounting for only 17.6%, and was concentrated in the 26–45 cohorts. Lack of awareness emerged as an important perceived contributor among younger adults, especially those aged 18–25, representing 40.2% of this group.

### 3.5. Practices and Behaviors Related to Pollution

Figure 5 highlights both the protective practices used by officers and their beliefs about the effectiveness of those measures. Wearing a mask was identified as the most important measure, mentioned by 76 respondents (63.3%). Among these, 30 officers (39.5%) believed masks were moderately effective, 22 (28.9%) considered them effective, 15 (19.7%) rated them as slightly effective, and 9 (11.8%) reported them as not effective. The survey did not differentiate between different types of masks, as the focus was general use and perceived effectiveness. Face covers or protective glasses were mentioned by 16 respondents (13.3%), with perceptions of effectiveness ranging from slightly effective (7 officers, 43.8%) to effective (5 officers, 31.3%), and 4 officers (25%) considered them moderately effective. Moving away from highly polluted areas was considered important by 72 officers (60%), though beliefs about effectiveness were more evenly distributed: 16 officers (22.2%) rated it as not effective, 19 (26.4%) slightly effective, 18 (25%) moderately effective, and 19 (26.4%) effective. A small number of respondents (n = 7, 5.8%) did not mention any protective methods, yet most still acknowledged the general importance of air pollution protection.

Despite this awareness, actual usage frequency was limited: most officers reported using protective methods sometimes (97 officers, 80.8%), with only 6 (5%) using them often and 4 (3.3%) always. A small number of respondents (n = 7, 5.8%) did not mention any protective methods, highlighting a gap between awareness of protective measures and consistent practical adoption. During the air quality monitoring around traffic police station, no officer was observed wearing masks or face-covering glasses except for helmets. Sometimes officers moved away from their fixed station to attend to a different task, perhaps in an effort to avoid high-exposure areas. When asked why they did not consistently use protective methods, many officers explained that the masks were uncomfortable and/or the masks restricted breathing or airflow during long working hours. Others reported affordability as a barrier, noting that purchasing protective masks on a consistent basis was financially difficult.

Future research should aim to quantify the objective effectiveness of these methods and explore factors influencing individual perceptions and compliance, such as accessibility, cost, comfort, and awareness campaigns [50]. Understanding both perceived and actual effectiveness can guide public health strategies to reduce exposure and improve health outcomes in polluted environments.

When asked about their annual health status check-up practice, the majority of traffic police officers (61.7%) reported attending medical check-ups occasionally (Table 7). Meanwhile 17.5% indicated that they attended such check-ups rarely, and only 14.2% reported undergoing check-ups regularly. A smaller proportion (6.7%) stated that they never had medical check-ups. This suggests that there is little health monitoring among traffic police officers, which could make it more difficult to identify and prevent occupational health issues early on and lowers their awareness of their own health status.

When respondents were asked which pollution-reduction measures they considered most effective, strategies targeting high-emission vehicles were prioritized. Restricting old or high-polluting vehicles was the most frequently cited measure (36.0% of responses; 49.6% of cases), reflecting strong agreement that phasing-out outdated vehicles could substantially improve air quality. This is consistent with old vehicles being perceived as a significant pollution source (Figure 4). Vehicle restriction systems, such as odd–even number plate policies, were also recognized (20.5% of responses; 28.2% of cases) as viable strategies. In contrast, preventing cars from idling at intersections (3.7% of responses; 5.1% of cases) and restricting traffic around sensitive areas (12.4% of responses; 17.1% of cases) were less favored, while a considerable portion of respondents (27.3% of responses; 37.6% of cases) selected “none of the above,” suggesting either skepticism regarding the listed measures or preference for alternative approaches. Overall, vehicle regulation—particularly targeting high-emission vehicles—was perceived as the most effective method to reduce workplace pollution.

### 3.6. Pollution Exposure and Officer Health

A cross-tabulation analysis was conducted to assess the distribution of self-reported health symptoms across four categories of perceived air pollution levels at the workplace (very high, *n* = 31; high, *n* = 39; moderate, *n* = 43; other (don’t know), *n* = 7) among traffic police officers. The analysis revealed several significant associations between perceived exposure levels and health outcomes, as shown in Table 8.

Statistically significant associations were observed between reported pollution level at the workplace and several health outcomes. A strong association was found with eye irritation (*p* < 0.0010). The prevalence of eye irritation increases markedly with higher pollution levels, rising from 25.8% in very-high-pollution areas to 74.2% in high-pollution areas, and it was most prevalent (79.1%) in moderate-pollution areas.

Significant associations were observed between pollution levels and several health outcomes. Sneezing (*p* = 0.009) was most frequent in the moderate (76.7%) and very high (58.1%) pollution groups. Runny nose (*p* = 0.011) was common in both very high and moderate groups (68.8%), while sore throat (*p* = 0.013) peaked in the high pollution group (74.4%). Psychological stress also showed a significant association (*p* = 0.021), reported by 44.2% of the moderate and 41.9% of the very high pollution groups. These results indicate that both respiratory symptoms and stress increase with elevated pollution exposure, em-phasizing the public health impacts of air pollution.

In contrast, cough prevalence ranged between 22 (71.0%) in the very high group, 29 (74.4%) in the high group, 36 (83.7%) in the moderate group, and 3 (42.9%) in the other group; however, this variation was not statistically significant (*p* = 0.118). Blood pressure disturbances were observed in 7 participants (22.6%) in the very high group, 3 (7.7%) in the high group, 10 (23.3%) in the moderate group, and 1 (14.3%) in the other group, with no statistically significant difference between groups (*p* = 0.242).

These findings demonstrate that significant occupational exposure levels of air pollution are associated with significantly greater rates of eye irritation, sneezing, runny nose, and psychological symptoms. The high prevalence of non-significant trends, such as those for cough and blood pressure, also points to a wider spectrum of potential health effects that may warrant further investigation with larger sample sizes. Consequently, these results argue compellingly for the implementation of targeted protective measures—such as the provision of high-grade personal protective equipment (e.g., N95 respirators and protective eyewear)—and for the inclusion of air quality mitigation strategies and mental health support within occupational safety protocols for traffic personnel. Future research should aim to correlate subjective health reports with objective environmental monitoring data to strengthen causal inference.

### 3.7. How Should Public Education on Air Pollution Be Delivered

To design an effective public education campaign on the impacts of air pollution, it is essential to understand the audience and tailor strategies accordingly. This involves many of the questions that were raised earlier, concerning the role of educational background in the understanding of air pollution and associated health impacts, the role of officers in environmental management and public education, and how their perceived role in mitigation is influenced by their understanding of health risks. Understanding information sources concerning air quality should also be understood for varying levels of awareness. By combining these insights and prioritizing channels that reach the largest and most relevant segments of the population, an education campaign can deliver targeted, inclusive, and impactful messages that raise awareness and encourage behavioral change to reduce air pollution and protect public health. Several questions are examined in this section: For those officers that are aware of AQ issues, where are they obtaining their information? For those that are unaware of AQ issues, what are their media consumption habits? From this, we can draw conclusions relating to the larger population.

The findings presented in Table 3 and Figure 6 highlight participants’ awareness, beliefs, and sources of information regarding air pollution. The analysis of pollution information source among traffic police officer across the four/three AQI awareness cohorts provides insights into how communication channels shape environmental awareness and behavioral engagement (Figure 6). 

Understanding where officers obtain AQI information and how this varies by awareness level also offers broader implications for designing public education on air pollution. Mass media was the most frequently reported information source for air pollution for traffic police officers, ranging from 66.7% among those who “know and follow AQI regularly”, 81.8% those who know but do not follow, 73.6% of the officers who know it exists but do not understand, and 75.0% of those who do not know but want to learn.

While mass media effectively raises general awareness, this does not translate into action or regularly following AQI for officers, indicating a gap in actionable guidance. Health workers emerged as more influential for fostering understanding and action, particularly among those who regularly follow AQI updates (25.0% each). Health workers influence 12.1% of those who know but do not follow, 18.9% of those who know it exists but do not understand, and only 5% of respondents who do not know but want to learn. This suggests untapped potential for targeted educational intervention.

The local environmental authority was reported by 25.0% of officers who know and follow AQI regularly, 21.2% who know but do not follow, 17.0% who know it exists but do not understand, and 25.0% of those who do not know but want to learn. This indicates that local authorities are a moderately trusted and visible source across all groups. Notably, they are equally represented among those who do not know but want to learn, suggesting their potential as an entry point for educating uninformed populations and increasing AQI comprehension. Governmental and non-governmental sources of information for air pollution were identified by 8.3% of officers who know and follow AQI, 15.2% who know but do not follow, 9.4% who know it exists but do not understand, and none of those who do not know but want to learn. These data suggest that institutional communication is relatively weak and does not effectively reach individuals who are less informed or seeking to learn about AQI. While these organizations are expected to be credible sources, their limited engagement reflects a potential gap in traffic police information campaigns or accessibility of AQI education.

## 4. Discussion

The study assessed knowledge, perceptions, and practice related to air pollution among traffic police officers in Addis Ababa. The findings revealed substantial exposure to air pollutants and a high prevalence of respiratory and eye symptoms. Remarkably, all officers reported being non-smokers, despite the national smoking prevalence in Ethiopia being low but not zero [49]. This finding highlights that tobacco use is unlikely to confound the observed health effects. Notably, 75% of the traffic police reported experiencing cough, suggesting that these respiratory symptoms are more plausibly attributable to occupational-exposure-/traffic-related air pollution rather than personal behavior.

This study also found that while most respondents were aware of AQI, understanding and active engagement were limited. Like previous research [51], many participants knew AQI exists but did not understand its significance or regularly monitor it (see Table 3 and Figure 3), reflecting the broader trend that awareness does not always translate into trust or use of official air quality data. Additionally, our findings align with the observations of Huang and Lee [52], suggesting that although higher education can improve knowledge, behavioral adoption of protective measures is influenced by attitudes, perceived risk, and barriers to adoption. Together, these results highlight the need for targeted educational intervention and accessible AQI communication to promote proactive engagement with air quality. Since traffic officers often engage with the public and enforce laws, increasing AQ knowledge among them would likely result in increased public awareness.

Knowledge about air pollution and related governmental pollution policies among traffic police officers was generally low (see Table 3 and Figure 3). Awareness of AQI was limited (10%), and a substantial proportion of officers were unaware of governmental initiatives targeting air pollution (54.2%). This limited awareness may stem from a lack of training (75.8% of officers reported having received no formal training on air pollution), limited collaboration with concerned institutions, and the fact that most officers relied on informal information sources, such as mass media, which has been shown to have a dominant role in disseminating pollution-related information.

Overall, the results highlight that mass media is the most widely accessed source, but it does not necessarily promote understanding or regular following of AQI. This pattern indicates that while broad channels like mass media effectively raise general awareness, they are limited in promoting consistent behavioral change, as information delivered through impersonal sources often remains at the recognition level rather than translating into protective action [53,54]. Health workers and local environmental authorities have targeted influence, particularly among those who already know about AQI or are motivated to learn.

When connected to the broader question of how public education on air pollution should be delivered, these findings suggest the most effective means is a tiered communication strategy that mirrors the progression of AQI awareness. Campaigns must segment the audience, as McCarron and Semple [55] demonstrate that simplified, visual materials enhance comprehension for those with lower educational attainment, while detailed, systemic content better engages highly educated individuals, potentially equipping them as peer educators [55,56]. Furthermore, awareness must be linked to tangible health outcomes, as Cibin and Horgan [56] and Wang and Cao [57] found that risk perception of personal health threats is a critical mediator motivating protective behaviors. Tools such as the AQI cannot remain abstract metrics; they must be translated into clear, actionable protocols to bridge the well-documented knowledge–behavior gap [56]. Finally, this education must be reinforced through mandatory training programs and trusted institutional channels, as consistent messaging from authoritative sources is essential for building the trust required to override inconsistent informal messaging. By integrating these insights, public education can transition from a generic model to a strategically segmented approach that effectively protects high-risk groups.

This finding appears to be in contrast with traffic officers in Kenya, where 96.3% indicated they had knowledge of the laws on traffic-related pollution and traffic regulations training. A total of 98.1% had attended a traffic management course. A total of 76.6% indicated that they had useful motor vehicle air pollution information. This suggests that stronger institutional exposure to regulatory frameworks is achievable [36].

In our study, traffic police officers in Addis Ababa consistently identified vehicle emissions (73.3%) as the primary contributor to outdoor air pollution. Other sources were reported less frequently, including construction (35.0%), industrial activities (32.5%), garbage/waste disposal (30.8%), and cooking (15.8%). Similarly, traffic police officers in Kampala demonstrated good knowledge of the sources and health effects of outdoor air pollution, which is consistent with our study in Addis Ababa, where officers also identified major pollution sources and associated health impacts [58]. Perceptions of officers align closely with empirical evidence from Addis Ababa, where vehicular sources (28%), biomass burning (18.3%), and oil dust (17.4%) comprise about two-thirds of the PM_2.5_ mass, followed by sulfate (6.5%) [12].

This study also provides population-level insights into perceived causes of traffic-related air pollution in Addis Ababa, extrapolated from traffic police assessments. Old vehicles are viewed as the most pressing issue, followed by congestion and, to a lesser extent, lack of awareness and poor traffic light management. These perception-based findings are consistent with investigations measuring traffic-related air pollution at intersections in Addis Ababa. They demonstrated that during severe congestion events, the concentration of PM increased, and they found that traffic volume, the heavy vehicle share, and traffic signal timing patterns play critical roles in pollutant concentration levels [11,59]. This evidence supports the extrapolated general perception that old vehicles and congestion are dominant contributors to air pollution in the city while also confirming that poor traffic management worsens pollutant emissions during peak flow conditions. This aligns with existing report on Addis Ababa’s reliance on outdated and poorly maintained cars, which contribute disproportionality to urban air pollution [60].

Regarding pollution trends, more than half of the officers believed that air quality has improved in recent years, although a notable portion perceived it as worsening or unchanged, reflecting ongoing challenges in urban pollution control. Similarly, officers reported varying health impacts from pollution exposure, ranging from mild to severe, indicating that perceptions of pollution are closely linked to experienced health outcomes.

The high prevalence of respiratory and eye symptoms emphasizes that traffic policing constitutes a high-risk occupation in megacities of developing countries. The observed psychological symptoms may reflect stress and fatigue resulting from both environmental exposure and occupational demands. Studies in Dhaka and Nairobi [36,48] have shown that traffic police are generally aware of motor vehicle emissions and related regulations, yet their daily duties expose them to significant health risks. Findings indicate that prolonged exposure to air pollution is associated with respiratory, cardiovascular, and eye problems and continues to negatively affect the wellbeing and performance of officers despite their awareness [12,61,62]. Routine health monitoring, targeted interventions such as the provision of personal protective equipment (e.g., masks with particulate filtration), and workplace modifications are essential to mitigate these risks. These results highlight the importance of preventive measures, protective equipment, and policy interventions to safeguard traffic police from the harmful effects of vehicular emissions.

When discussing challenges for pollution reduction, officers pointed to systemic barriers, including lack of coordination among institutions, insufficient legal support, limited training and knowledge, inadequate public cooperation, and the absence of advanced monitoring technologies. They emphasized the need to strengthen AQI education through targeted campaigns and accessible media platforms to improve understanding of air quality issues and promote behavioral change.

This study has limitations. First, it did not include objective health assessments, such as biomarkers for objectively assessing non-smoking status (e.g., cotinine) or clinical examinations including spirometry, which would have provided more accurate, objective, and quantitative health data. These assessments were not feasible due to financial, temporal, logistical, and security constraints among traffic police personnel. Second, the study relied on self-reported data, which may introduce reporting bias. Third, the absence of a control group (i.e., officers not exposed to high levels of traffic pollution) and the relatively small sample size limit the generalizability of the findings. Despite these limitations, the study provides valuable insights into traffic police knowledge, perceptions, and occupational health risks. Future research should incorporate objective exposure and health assessments, larger sample sizes, and appropriate comparison groups to better quantify exposure-related health effects.

## 5. Conclusions and Recommendations

This study reveals that traffic police officers in Addis Ababa have limited knowledge, suboptimal perception, and inadequate practices regarding occupational exposure to air pollution. Despite the availability of monitoring tools such as the Air Quality Index, officers underutilize them due to insufficient training and guidance. These gaps underscore the need for targeted interventions, including comprehensive education on air pollution health impacts, proper use of protective equipment, and institutional support such as duty rotation and provision of high-filtration respirators. Addressing these issues is essential to safeguard the health of officers and improve occupational safety policies in urban traffic management.

## Figures and Tables

**Figure 1 ijerph-23-00060-f001:**
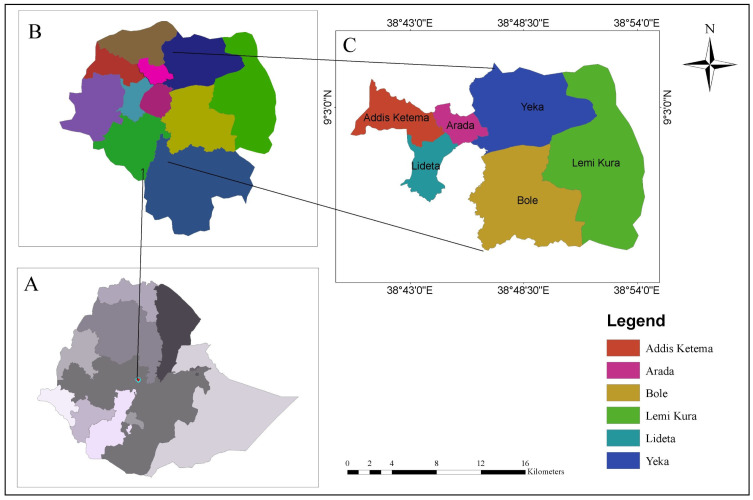
Study area map: (**A**) Political boundaries of Ethiopia showing regions and the Addis Ababa City Administration; (**B**) Administrative boundaries of Addis Ababa and its eleven sub-cities; (**C**) Sub-cities of Addis Ababa where study participants were recruited.

**Figure 2 ijerph-23-00060-f002:**
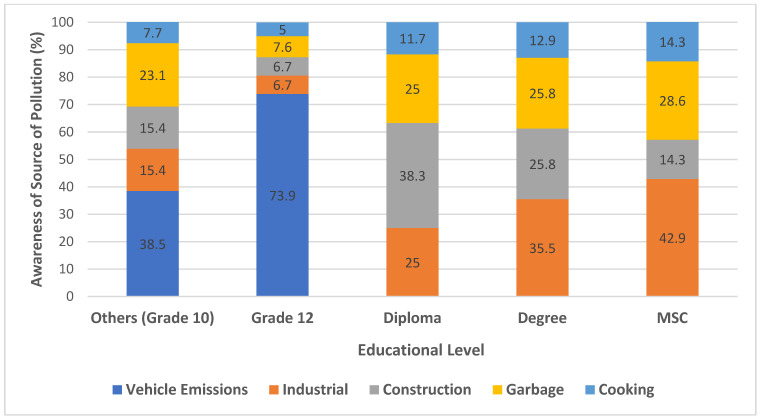
Air pollution source awareness by educational level among traffic police. This figure illustrates the percentage of awareness regarding different sources of pollution among individuals with varying educational backgrounds.

**Figure 3 ijerph-23-00060-f003:**
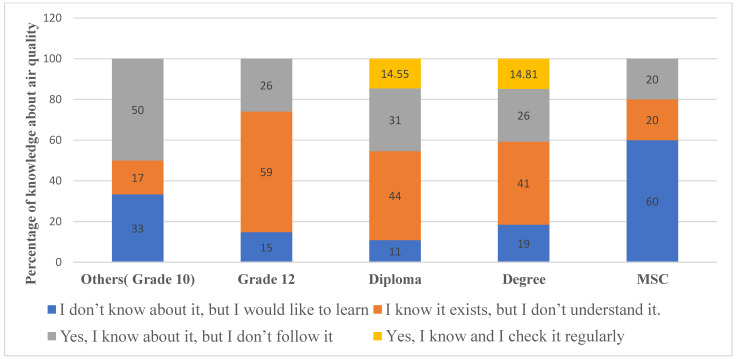
Assessment of traffic police awareness of the Air Quality Index (AQI) across educational levels.

**Figure 4 ijerph-23-00060-f004:**
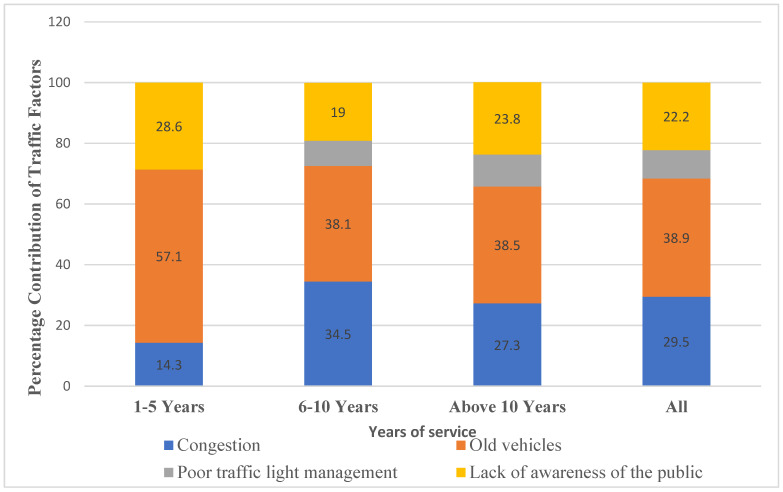
Distribution of respondents’ perceptions of air pollution causes across years of service of traffic police experience. The bar graph shows respondents with different years of work experience perceive the main causes of pollution.

**Figure 5 ijerph-23-00060-f005:**
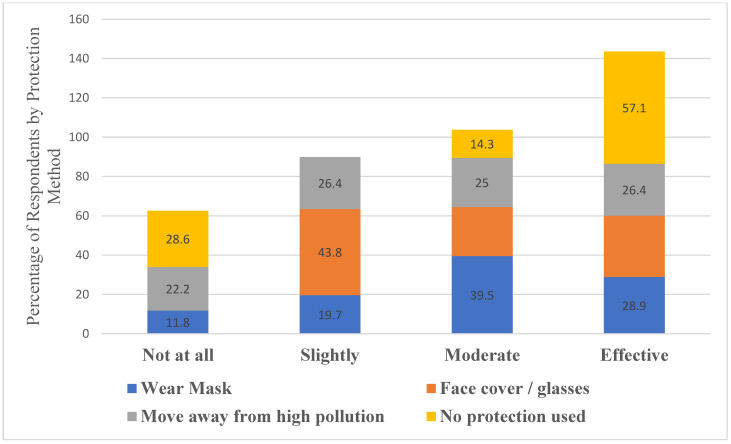
Distribution of protective measures used by traffic police across beliefs in the effectiveness of air pollution protection methods. The bars represent the proportion of respondents within each belief category who reported using each protective measure.

**Figure 6 ijerph-23-00060-f006:**
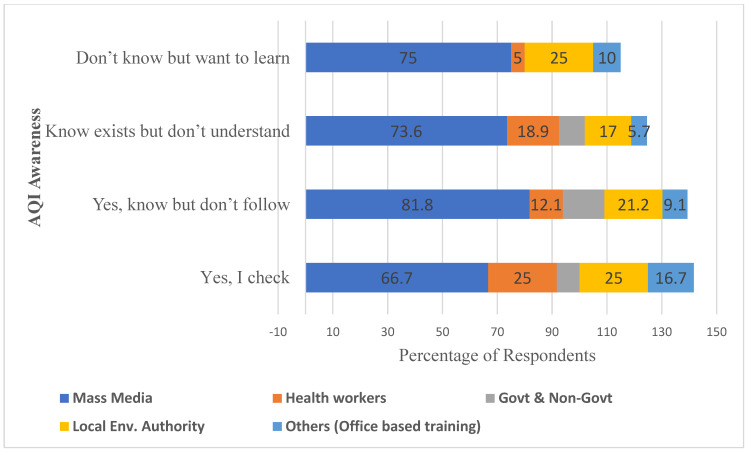
Distribution of sources of air pollution information among traffic police across different levels of awareness of AQI. Bars represent the proportion of respondents who obtain information from each source.

**Table 1 ijerph-23-00060-t001:** Demographic characteristics of the respondents.

Category	Group	Frequency	Percent (%)
Gender	Male	96	80.0
	Female	24	20.0
Marital Status	Married	112	93.3
	Single	7	5.8
	Divorced	1	0.9
Age Group	18–25	6	5.0
	26–35	46	38.3
	36–45	58	48.3
	46 and Above	10	8.3
Service Year	1–5 Years	6	5.0
	6–10 Years	39	32.5
	Above 10 Years	75	62.5
Educational Level	Grade 12	27	22.5
	Diploma	55	45.8
	Degree	27	22.5
	MSC	5	4.2
	Others (Grade 10)	6	5
Cigarette Smoking	No	120	100
Alcohol Use	Yes	21	17.5
	No	99	82.5
Frequency of Drinking Alcohol	Frequently	2	1.7
	Occasionally	19	15.8
Total		120	100

**Table 2 ijerph-23-00060-t002:** Prevalence of reported symptoms among participants (*n* = 120).

Symptom	Yes, n (%)	No, n (%)	Total, n	Frequency Detail *, n (%)
High blood pressure	21 (17.5)	99 (82.5)	120	–
Sore throat	35 (29.2)	85 (70.8)	120	Rarely: 29 (24.2), Frequently: 5 (4.2)
Runny nose	67 (55.8)	53 (44.2)	120	–
Sneezing	70 (58.3)	50 (41.7)	120	Rarely: 59 (49.2), Frequently: 11 (15.7)
Eye irritation	74 (61.7)	46 (38.3)	120	Rarely: 64 (53.3), Frequently: 12 (10)
Cough	90 (75.0)	30 (25.0)	120	Rarely: 70 (58.3), Frequently: 20 (16.7)
Difficulty focusing/concentrating	18 (15.0)	102 (85.0)	120	–
Psychological stress	40 (33.3)	80 (66.7)	120	–
Sleep loss	32 (26.7)	88 (73.3)	120	–

* Frequency detail refers to the reported occurrence of symptoms among participants who answered “Yes”.

**Table 3 ijerph-23-00060-t003:** Participant pollution-related training, policy awareness, and AQI knowledge (*n* = 120).

Item	Response Category	n (%)
Received pollution training	Yes	29 (24.2)
	No	91 (75.8)
Awareness of government pollution reduction policy	Yes	55 (45.8)
	No	65 (54.2)
Collaboration with environmental protection authorities	Yes regular	11 (9.1)
	No	29 (24.2)
	Occasionally	45 (37.5)
	Don’t know	35 (29.2)
Awareness and understanding of AQI	Yes, I know and I check it regularly	12 (10)
	Yes, I know but I don’t follow it	35 (29.2)
	Yes, I know but I don’t understand it	53 (44.2)
	I don’t know but I would like to learn	20 (16.7)
	Total	120 (100.0)

**Table 4 ijerph-23-00060-t004:** Distribution of awareness among traffic police on air pollution health risks, showing perceived health risks by age group (multiple responses).

Health Risk	18–25	26–35	36–45	46+	Total
Respiratory problems	5 (6.10%)	28 (34.15%)	39 (47.56%)	10 (12.20%)	82
Heart-related disease	1 (2.94%)	15 (44.12%)	17 (50.00%)	1 (2.94%)	34
Increased blood pressure	1 (2.00%)	18 (36.00%)	27 (54.00%)	4 (8.00%)	50
Lung-related damage	1 (2.78%)	16 (44.44%)	17 (47.22%)	2 (5.56%)	36
Did not know	0 (0.00%)	0 (0.00%)	1 (100.0%)	0 (0.00%)	1
**Total**	6 (5.08%)	45 (38.14%)	57 (48.31%)	10 (8.47%)	118

**Table 5 ijerph-23-00060-t005:** Summary of respondents’ perception regarding workplace air pollution level, health impact, and pollution trend (N = 120).

Variable	Category	N (%)	Cumulative (%)
Perceived Pollution Level at Workplace	Very High	31 (25.8)	25.8
	High	39 (32.5)	58.3
	Moderate	43 (35.8)	94.2
	Other (Do Not Know)	7 (5.8)	100.0
Health Impact of Pollution on Officers	None (No Impact)	23 (19.2)	19.2
	Mild	45 (37.5)	56.7
	Severe	46 (38.3)	95.0
	Do Not Know	6 (5.0)	100.0
Pollution Quality in Recent Years	Improved	62 (51.7)	51.7
	Worsened	35 (29.2)	80.8
	No Change	15 (12.5)	93.3
	Do Not Know	8 (6.7)	100.0
Challenges for Pollution Reduction	Lack of Coordination	26 (21.7)	21.7
	Lack of Advanced Pollution Monitoring Technology	69 (57.5)	57.5
	Low Public Cooperation	46 (38.3)	38.3
	Insufficient Legal Support	51 (42.5)	42.5
	Lack of Adequate Training/Knowledge	56 (46.7)	46.7

**Table 6 ijerph-23-00060-t006:** Extrapolated estimates of traffic issues in the Addis Ababa population by vehicle age group.

Age Group	Congestion	Old Vehicles	Poor Traffic Light	Lack of Awareness	Total Population
18–25	310,610 (50.0%)	414,147 (66.7%)	103,537 (16.7%)	207,073 (33.3%)	621,220
26–35	466,416 (58.7%)	500,965 (63.0%)	207,296 (26.1%)	328,218 (41.3%)	794,634
36–45	339,491 (60.3%)	484,987 (86.2%)	77,598 (13.8%)	261,893 (46.6%)	562,585
46+	207,577 (40.0%)	415,154 (80.0%)	51,894 (10.0%)	207,577 (40.0%)	518,942
All	1,324,094 (53.0%)	1,815,253 (72.7%)	440,325 (17.6%)	1,004,761 (40.2%)	2,497,381

**Table 7 ijerph-23-00060-t007:** Practices and behaviors related to pollution frequency of respondents’ medical check-ups and pollution reduction measures.

Variable	Category	F (%)	Cumulative (%)
Regular Medical Check-ups	Often	17 (14.2)	14.2
	Occasionally	74 (61.7)	75.8
	Rarely	21 (17.5)	93.3
	Never	8 (6.7)	100.0
Measures to Reduce Pollution in Workplace, None of the Above	Vehicle restriction	33 (27.5)	27.5
	Restricting old/high-emission vehicles	58 (48.3)	48.3
	Preventing cars from idling	6 (5.0)	5.0
	Restricting traffic around schools/hospitals	20 (16.7)	16.7
	None of the above are practiced	44 (36.7)	36.7

**Table 8 ijerph-23-00060-t008:** Association between self-reported pollution exposure level and prevalence of health symptoms among study participants (N = 120).

Pollution Level	n	BP Yes (%)	S Yes (%)	RN Yes (%)	Sneezing Yes (%)	I Yes (%)	Cough Yes (%)	DF Yes (%)	PS Yes (%)	SL Yes (%)
Very High	31	7 (22.6)	15 (48.4)	20 (64.5)	18 (58.1)	23 (74.2)	22 (71.0)	9 (29.0)	13 (41.9)	10 (32.3)
High	39	3 (7.7)	7 (17.9)	15 (38.5)	16 (41.0)	13 (33.3)	29 (74.4)	5 (12.8)	8 (20.5)	7 (17.9)
Moderate	43	10 (23.3)	13 (30.2)	30 (69.8)	33 (76.7)	34 (79.1)	36 (83.7)	4 (9.3)	19 (44.2)	15 (34.9)
Other	7	1 (14.3)	0 (0.0)	2 (28.6)	3 (42.9)	4 (57.1)	3 (42.9)	0 (0.0)	0 (0.0)	0 (0.0)
*p*-value		0.242	0.013	0.011	0.009	<0.001	0.118	0.064	0.021	0.110

BP = high blood pressure; S = sore throat; RN = runny nose; I = eye irritation; Cough = coughing; DF = difficulty focusing; PS = psychological stress; SL = sleep loss; “Yes (%)” indicates the percentage of respondents who reported experiencing the symptom.

## Data Availability

The original contributions presented in this study are included in the article and Appendix A (Survey Questions). Further inquiries can be directed to the corresponding author.

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
