# Peer review of "Knowledge, Perceptions, and Practices of Traffic Police Officers Towards Air Pollution in Addis Ababa, Ethiopia: An Exploratory Study"

_ijerph, 2025, doi:10.3390/ijerph23010060_

Round 1

Reviewer 1 Report

Comments and Suggestions for Authors

This study explored occupational exposure, protective practices, and health risks and perceptions and awareness of air quality associated health risks among 120 traffic police officers in Addis Ababa.

1 For paragraph 1, the data should be updated.

2 For the Introduction, it is not enough to state that there is no study in Addis Ababa. The authors should point out the limitations of current studies.

3 Lines 165-167, the explanation of parameters should be broken off with commas.

4 The title of the coordinate axis should be included in figures 2-7.

5 There are two figures 4 in this version, please revise them.

Reviewer 2 Report

Comments and Suggestions for Authors

Well-researched study and well-written manuscript. I believe it makes a very good contribution to the issue of how best to approach training and respiratory protection of workers who are exposed continually to air pollution as a part of their job.

I have only a few suggestions: (1) in Section 2.1 the authors might consider using Fahrenheit scale in parentheses after the temperatures are provided in Centigrade just to increase reader understanding; (2) the finding that the sample population had no smokers (presumably tobacco smokers) is interesting in that smoking prevalence in Ethiopia is low but not zero. See Mengesha SD et al at https://doi.org/10.1186/s12889-022-12893-8 . This finding makes the 75% prevalence of cough even more likely to be occupational in origin. Perhaps drawing the reader's attention to this particular finding might be helpful; (3) the term "masks" is used but I could not find a description of what kind of "masks" the survey participants used, e.g., are the masks actually a N95 respirator or something high-grade like an N95 filtering facepiece respirator or were the masks cloth type facial coverings without much air filtering capacity?

Reviewer 3 Report

Comments and Suggestions for Authors

A brief overview of the study

The study examined the knowledge, perceptions, and practices of traffic police officers regarding air pollution and found high exposure to harmful particles, along with frequent respiratory and eye symptoms. The officers demonstrated poor knowledge of governmental policies and the Air Quality Index (AQI), partly due to a lack of formal training and reliance on mass media as their primary source of information. Although the media increase general awareness, they rarely lead to behavioral change, whereas health workers and environmental institutions have a stronger educational impact. The findings suggest that campaigns should be tailored to different knowledge levels and that AQI information must be incorporated into practical guidelines to bridge the gap between awareness and actual protective behavior. Numerous health burdens were reported, including respiratory, cardiovascular, and psychological problems, consistent with findings from other countries. Participants felt that improvements in air quality were slow and uneven, with major systemic barriers such as weak institutional coordination, limited training, and the absence of technological monitoring. Recommendations include mandatory education on air pollution, proper use of protective equipment, and better understanding of the AQI. Introducing regular health screenings, providing high-quality respirators, and rotating officers through less polluted posts are also important. Overall, the study emphasizes the need for stronger institutional support and targeted public health measures to protect this high-risk occupational group.

Advantages

The results of the study are valuable because traffic police officers represent a group that is directly and almost daily exposed to high levels of motor-vehicle exhaust emissions. Therefore, the study provides relevant and practical insights into how an exposed population perceives pollution. By examining subjective attitudes and self-reported health complaints, the study offers a broader understanding of the specific effects of traffic-related pollution on traffic police officers. It highlights deficiencies in basic training, weak institutional coordination, and poor information flow, which serve as a solid foundation for policy development and training improvements. Comparison with findings from other countries gives the study broader context. The results also lead to measures that can be directly applied in practice, such as improving training quality, job rotation, and the use of personal protective equipment. All study participants identified themselves as non-smokers. Since smoking is one of the major risk factors for respiratory and cardiovascular problems, the absence of smokers simplifies attribution of the observed symptoms to air pollution, reducing the likelihood that the health issues are caused by personal habits. This strengthens the validity and reliability of the conclusions.

Disadvantages

Despite the strengths of the study, several limitations can be identified. First, perceptions and self-reported symptoms are subjective and prone to bias. A major limitation is the absence of actual measurements of pollutant concentrations, which would allow comparison between subjective experience and real exposure levels. The fact that all participants were non-smokers, although previously mentioned as an advantage, can also be considered a disadvantage. The sample may be unrepresentative, since a significant portion of police officers are smokers; therefore, the study may not reflect the characteristics of the broader police population. The ability to compare findings with other studies is also reduced, as most similar research includes both smokers and non-smokers. Additionally, the study cannot assess interactions between smoking and pollution—namely, whether smokers experience weaker or stronger symptoms.

Recommendations

The overall conclusion is that the study lacks sufficient objectivity, which substantially limits the applicability of its findings. The study could be improved through several methodological and operational enhancements. First, objective exposure measurements should be included, such as personal PM2.5 and PM10 monitors, fixed samplers at major intersections, or the use of local monitoring station data, in order to compare subjective assessments with actual pollutant concentrations in the work environment. Although all participants declared themselves non-smokers, the study should verify non-smoking status using biomarkers such as cotinine to reduce the risk of misclassification. Basic medical examinations, including spirometry, could also be incorporated to provide objective health assessments.

Further improvement can be achieved by expanding the sample size and introducing a control group (e.g., police officers working in office settings), which would enable more accurate comparisons of the health effects associated with exposure to traffic-related air pollution.

Reviewer 4 Report

Comments and Suggestions for Authors

I am sorry since the topic is really interesting but the author should revise in depth the paper before submitting it for publication and in the present form it cannot be accepted.

-Line 147: fig 1 need to revise as more legible.

Section 2.2, section 3.4 and section 4 absent! be careful'

-Section 2.2. Data section lacks of in-depth analysis and novelty in evaluation methodology. I think requires the manuscript should contain sufficient contributions to the new body of knowledge.
Line 241-249: Need to improve research skills as this paper does not meet the minimum standard requirements in terms of reviewing the related literature and previous studies, are not identified/clear, not using of any software for analysis, the discussion/conclusion is rather general and not answering the research questions/implications etc.

Line 185-209: The results and the discussion can go together so that you can have a quick understanding of the analysis performed.

Line 629-640: Please enter the conclusions of your work in the corresponding place

-Also all figs need to revise e.g 100.0% in fig2?

Round 2

Reviewer 3 Report

Comments and Suggestions for Authors

I have no further comments.

Reviewer 4 Report

Comments and Suggestions for Authors

its suitable for publication, author well prepared and current form ist ok for publishing.